# On Health Effects of Resveratrol in Wine

**DOI:** 10.3390/ijerph19053110

**Published:** 2022-03-06

**Authors:** Robin Haunschild, Werner Marx

**Affiliations:** Max Planck Institute for Solid State Research, Heisenbergstr. 1, 70569 Stuttgart, Germany; w.marx@fkf.mpg.de

**Keywords:** scientometrics, bibliometric analysis, resveratrol, red wine, health effects, RPYS

## Abstract

We analyzed 3344 publications concerned with the health-related effects of resveratrol that occurs in wine and grapes. We discovered that publication activity increased until 2010 and decreased slightly afterwards. The most frequent author keywords were classified into six groups: (1) beverage-related keywords, (2) compound-related keywords, (3) disease-related keywords, (4) effect-related keywords, (5) mechanism-related keywords, and (6) broader keywords. By means of reference publication year spectroscopy, we analyzed and discussed the most frequently cited references (i.e., key papers) within the publication set. A rather large portion of the key papers exhibit a deliberative or positive attitude and report on the health effects of resveratrol, although limited data in humans preclude drawing unambiguous conclusions on its health-related benefits. From our analysis, we could not identify specific publications that provide a distinct change of direction of the ongoing scientific discourse. Moderate red wine consumption seems to bear the potential of being health promoting, whereas excessive alcohol consumption can induce liver cirrhosis and cancer.

## 1. Introduction

Wine is a complex beverage, containing a wide range of polyphenols, which represent the main class of bioactive compounds. Polyphenols are considered to affect health by preventing atherosclerosis and coronary heart disease (CHD) and are seen as potential cancer chemo-preventive agents. The content and profile of polyphenols depend on wine grape variety, geographical origin, growing methods, and vinification processes [1]. Red wines usually contain an order of magnitude more polyphenols than white wines [2]. Among the broad compound class of polyphenols, resveratrol and its derivatives play a major role. Resveratrol (trans-3,5,4′-trihydoxystilbene) was first isolated in 1939/40 by Michio Takaoka [3,4] as a constituent of the roots of the medical plant white hellebore (Veratrum grandiflorum). We use the terms resveratrol and trans-resveratrol interchangeably.

Since about 1990, the possible benefits of moderate consumption of red wine in the prevention of cancer and heart diseases have received increasing attention in the scientific community and in the popular media as well. The cardioprotective effects and the ability to suppress the proliferation of a wide variety of tumor cells made resveratrol a promising drug in human medicine [5]. According to Wu, et al. [6], such attention was prompted by research findings in the 1980s, supporting a relationship between red wine consumption and the French paradox [7,8]: Although the French diet was richer in saturated fats and the concentration of plasma cholesterol was much higher, the mortality from cardiovascular disease was comparatively low. The high consumption of red wine during meals was suggested as a possible explanation. This issue is still a topic among scientific discussions. Recently, Zhang, et al. [9] reviewed the pharmacological literature regarding resveratrol and called for further research.

Until now, there is only one bibliometric analysis dealing with the newly emerging research topic about possible health effects of polyphenols and in particular of resveratrol in red wines [2]. This paper provides a comprehensive overview of the relevant literature with regard to the contributing authors, the origin of the major contributors and their institutions, the type of papers, and the research themes. However, the study does not present an in-depth analysis based on citation (impact) data, although a time curve of the overall citations of the publication set analyzed is presented.

It would be interesting for many people to acquire knowledge regarding the current status of discourse: Is there convincing evidence for health effects of resveratrol and particularly of moderate red wine consumption? What do the in vivo studies based on humans reveal? Which kind of research should be supported further to obtain a clearer picture? We assume that the publications of the experts dealing with this research topic, which were most frequently cited within the research community, can deliver some interesting information. However, an ordinary citation ranking is by far insufficient. We need to compare the citation impact of each paper with similar papers (i.e., papers of the same research topic and publication year).

This study aims to reveal the key papers of the scientific discourse concerning the relationship between resveratrol as a specific polyphenol in wine and health effects, based on an analysis of the most-cited papers (cited references) within a carefully selected publication set. This approach of citation analysis is based on what Galton at the beginning of the 20th century called the wisdom of crowds [10]. Citations measure the scientific impact of publications, which is one aspect of scientific quality. Revealing the key papers using citation analysis might deliver some information about the evolution of the scientific discourse. In consideration of the complexity of the underlying biochemistry and medicine, there is no easy path to gain a valid picture of the current status of the knowledge. In this study, we try to answer the following question using well-proofed bibliometric methods: What are the main topics and the research results of the papers, which are referenced (cited) most often within the relevant literature of the scientific community?

A bibliometric method called Reference Publication Year Spectroscopy (RPYS) [11,12] has been particularly developed to reveal the historical roots or origins of research fields. However, this method can also be used to identify the key papers within the continuing scientific discourse within such fields or topics. RPYS changes the perspective of citation analysis from a times cited to a cited reference analysis [13] and assumes that researchers produce useful data output by the references cited in their publications. These references can be analyzed statistically with regard to the publications most relevant for their specific field of research. Whereas single scientists judge the origins of their research field more or less subjectively, the overall community might deliver a more objective picture [14]. Researchers active in the field effectively “vote” by their cited references, to identify the publications most important for the evolution of their research field. This novel approach may at least partially avoid some of the subjectivity associated with the comparative evaluations of individual research studies in narrative reviews, now increasingly considered as a significant flaw within the peer review system.

RPYS utilizes the following observation: Analysis of the publication years of the references cited by all papers in a specific research field (i.e., analysis of the reference publication years, RPYs) shows that early RPYs are not equally represented. Some RPYs occur more frequently among the cited references than others. These RPYs appear as more or less pronounced peaks in the distribution of the RPYs (i.e., the RPY spectrogram). The peaks are frequently based on single or few publications, which have been referenced comparatively often compared to other publications of the relevant research community. These publications are of special significance to the analyzed research field.

This study aims to reveal those papers that have been cited comparatively often within the scientific discourse on health effects of the specific polyphenol resveratrol in wine. These key papers are analyzed further with regard to their topics and conclusions.

## 2. Dataset and Methods

### 2.1. Dataset

The topic of our research includes different disciplines (e.g., chemistry, medicine, biology, and life sciences). Therefore, we used the multidisciplinary database Web of Science (WoS) [15] for this study. In WoS, we employed the following search query to find publications related to resveratrol (probably also its derivatives) and wine: TS = (Resveratrol* SAME (Wine* OR Vin* OR Grap* OR Vitis*)). By this we searched for “resveratrol” and the wine synonyms in the title, abstract, and keywords of publications. The operator SAME ensures that the term resveratrol and any of the wine synonyms is found in the same topic field (e.g., both terms occur in the title, or both terms occur in one of the other two topic fields).

We analyzed the WoS subject categories within the result set to find health-related ones. We selected those health-related WoS subject categories to combine them with our search query related to resveratrol and wine for obtaining our final WoS query that is included in Appendix A. We retrieved 3344 papers on 5 October 2021.

### 2.2. Methods

We created co-occurrence maps of author keywords using VOSviewer [16]. The distance between two nodes is determined by the co-occurrence frequency of the terms. The size of the nodes is dependent on the number of publications with a specific keyword. The nodes can be colored according to various characteristics (e.g., cluster assignments, average number of citations, or average publication year). We colored the nodes according to the average publication years of the publications connected to them. We included author keywords if they occurred at least ten times.

We imported all 3344 papers with 181,218 cited references into CRExplorer http://crexplorer.net, (accessed on 8 February 2022); [17,18,19]. The distinct number of cited references amounts to 94,828. Up to 30 different citing publication years were observed. We applied the automatic algorithm for clustering equivalent cited references with volume and page activated and a Levenshtein threshold of 0.75. This enabled merging of 165 cited references. We removed 92,771 cited references that occurred less than ten times to sharpen the spectrogram and avoid noise. Thus, 1892 cited references remained for our analysis. The RPYS spectrogram was plotted using R [20] with the R package ‘BibPlots’ [21]. In addition to the static RPYS plot in this paper, we produced an interactive RPYS graph using the R package ‘dygraphs’ [22]. We analyzed the RPYS spectrogram regarding relevant peaks of the five-year median deviation. Tukey’s fences [23] were used to support the identification of the most important peaks: Important peaks are flagged based on the interquartile range of the median deviations [24]. If unflagged RPYs with a similar NCR value occur around flagged RPYs, those unflagged RPYs should also be investigated. In addition, cited references that were cited very frequently in many citing years (N_TOP10 ≥ 15) were analyzed [25]. Such very frequently cited references might not form pronounced peaks in the spectrogram. Thus, the N_TOP10 indicator is important to consider for finding frequently cited references. Moreover, we investigated the 15 most-referenced cited references (referenced at least 200 times).

## 3. Results

In this study, we included all document types (i.e., not only articles and reviews but also proceedings papers, meeting abstracts, book chapters, editorial materials, etc.) in our analysis. The vast majority of publications belong to the document types ‘article’ (*n* = 2735, 81.8%), ‘review’ (*n* = 447, 13.4%), and ‘proceedings paper’ (*n* = 131, 3.9%). Although some publications belong to more than one document type, these three groups contain 3200 (95.7%) of our analyzed publication set. In the following, we present three different analyses of our dataset: (1) An annual publication profile is presented. (2) Frequently occurring author keywords are analyzed. (3) The references cited in the publication set are analyzed and discussed.

### 3.1. Annual Publication Profile

Figure 1 shows the annual publication profile of papers that deal with resveratrol in connection with wine and health. The number of publications shows an overall increase until 2010 and slightly decreases afterwards.

### 3.2. Keyword Analysis

Figure 2 shows a semantic map of frequently occurring author keywords. Obvious synonyms (see Table A1 in the Appendix A) were grouped together. The nodes (i.e., keywords) are colored by the average publication year of the corresponding publications.

For an overview of the thematic content of the publications dealing with health effects of resveratrol in wine, we categorized the keywords as follows: (1) Keywords related to beverages containing resveratrol (e.g., studies that analyze the amount of resveratrol as a function of the wine grape variety, the geographical origin, and the vinification process). (2) Keywords related to specific chemical compounds, particularly to the biochemistry of polyphenols. (3) Keywords related to the possible prevention of specific diseases. (4) Keywords studying possible health effects of resveratrol in vitro or in vivo. (5) Keywords that are focused on the biochemical mechanism of a specific health effect, by which resveratrol possibly acts. We have assigned the keywords in Figure 2 to those topics and include a further category with unspecific or broader terms (each arranged in the order of occurrence):Beverages related: wine (133), red wine (108), grapes (72), vitis vinifera (37), grape juice (22), grape seed extract (18), grapevine (13), vitis amurensis (12), vitaceae (12).Compounds related: resveratrol (1531), polyphenols (306), stilbenes (90), nitric oxide (86), phenolics (57), flavonoids (56), reactive oxygen species (56), quercetin (54), alcohol (53), phytochemical (48), piceatannol (40), hplc (34), pterostilbene (34), curcumin (29), ethanol (26), egcg (21), cytokines (20), sirtuins (20), ampk (19), glutathione (16), phytoestrogen (15), red wine polyphenols (15), calcium (13), cytochrome p450 (13), stilbenoids (13), anthocyanins (12), caspase-3 (12), cholesterol (12), piceid (12), bcl-2 (11), catechin (11), cisplatin (11), cyclooxygenase (11), genistein (11), grape polyphenols (11), phytoalexin (11), superoxide dismutase (11), beta amyloid (10), doxorubicin (10), epsilon viniferin (10), hydrogen peroxide (10), lipopolysaccharide (10), melatonin (10), polydatin (10), polygonum cuspidatum (10).Diseases related: atherosclerosis (63), cardiovascular disease (63), cancer (58), alzheimer’s disease (46), diabetes (34), obesity (32), breast cancer (28), colon cancer (26), hypertension (25), liver (25), prostate cancer (23), diet (18), blood pressure (17), colorectal cancer (15), ischemia (15), metastasis (15), brain (14), heart (14), parkinson’s disease (14), nf2 (13), hippocampus (12), ischemial/reperfusion (12), cardiovascular (11), cerebral ischemia (11), neuroinflammation (11), microglia (10), neurodegeneration (10), osteoblast (10).Effects related: apoptosis (240), oxidative stress (201), antioxidant (189), inflammation (93), chemoprevention (68), neuroprotection (65), antioxidant activity (51), rats (41), cardioprotection (24), mediterranean diet (24), nutraceutical (23), anti-inflammatory (20), invasion (16), antioxidant capacity (13), prevention (13), cancer chemoprevention (12), clinical trials (12), health (12), anticancer (11), french paradox (11), neuroinflammation (11), high-fat diet (10), insulin resistance (10).Mechanisms related: cell cycle (39), sirt1 (55), nf-kappa b (38), angiogenesis (33), cytotoxicity (33), mitochondria (32), bioavailability (31), autophagy (28), metabolism (27), pharmacokinetics (27), endothelium (26), free radicals (26), p53 (25), lipid peroxidation (24), endothelial cells (22), platelets (16), gene expression (15), molecular docking (15), endothelial function (14), platelet aggregation (14), tnf-alpha (14), dna damage (13), macrophages (12), vegf (12), blood platelets (11), caco-2 cells (10), endothelial dysfunction (10), estrogen receptor (10), inducible nitric oxide synthase (10).Broader terms: aging (41), proliferation (28), akt (21), cell proliferation (20), longevity (12), migration (11), differentiation (10), natural products (10), ros (10), toxicity (10).

### 3.3. Reference Analysis

Figure 3 shows the RPYS spectrogram of the wine- and health-related resveratrol research. The grey bars represent the number of cited references (NCR) in each year, and the blue line shows the five-year median deviation. Outlier peaks are indicated by red stars and marked with their RPY.

Table 1 illustrates the most frequently cited references below the main peaks in Figure 3. Although 1939/1940 is not an outlier peak, we included CR1 and CR2 from these two years since they report about the first isolation of phenolic substances from the roots of the medical plant white hellebore (Veratrum grandiflorum). These CRs can be seen as the historical origin of research on resveratrol as a plant product.

In CR3, Langcake and Pryce reported the detection of trans-resveratrol in *Vitis Vinifera* and other *Vitaceaes* as a response to infection and injury. In CR4, Renaud and Delorgeril discussed the so-called French paradox for coronary heart disease (CHD) in connection with the consumption of alcohol and particularly of red wine. Inhibition of platelet aggregation by wine rather than an effect on atherosclerosis is presented as an explanation for protection from CHD in France. The paper presented the results of the study on the notion of “French paradox”. Serge Renaud is considered to be the father of the phrase. In CR5, Siemann and Creasy reported that the concentration of resveratrol in selected wines was determined as a function of wine grape variety, geographical origin, growing methods, and winemaking procedures. The results indicated that resveratrol might represent the active ingredient in wines causing reduction of serum lipids. In CR6 and CR7, Frankel et al. reported that besides the alcohol content of wine, the phenolic compounds in red wine are assumed to bear an important role in connection with the French paradox. The in vitro studies reveal that the antioxidant properties of the phenolic substances in red wine inhibit the copper-catalyzed oxidation of human LDL. In CR8, Fitzpatrick et al. investigated possible effects of various grape wines and other grape products on vascular function in vitro. They remarked that “if such responses occur in vivo, they could … contribute to a reduced incidence of coronary heart disease”. In CR9, Pace-Asciak et al. identified trans-resveratrol as the most important factor for the inhibition of platelet aggregation, thereby explaining the protective role of red wine against atherosclerosis and CHD. In CR10, Bertelli et al. examined the antiplatelet activity of synthetic and natural resveratrol in red wine. The results show that the antiaggregating effect of resveratrol is related to its concentration in red wine. In CR11, Jang et al. reported that resveratrol was found to exhibit cancer preventive activity by acting as an antioxidant and antimutagen. The data suggest that resveratrol merits investigations as a potential cancer chemopreventive agent in humans. In CR12, Gehm et al. presented a study that broadens the spectrum of the biological actions of phenolic compounds in grapes and wines. They examined the estrogenic actions of resveratrol as a phytoestrogen in connection with human breast cancer cells. In CR13, Soleas et al. reviewed the literature regarding resveratrol. The authors of this review miss conclusive evidence for the absorption of resveratrol by humans in biologically significant amounts. They question but do not exclude beneficial activities as a consequence of moderate red wine consumption.

In CR14, Fremont summarized the various biological effects of resveratrol isomers in connection with human cardiovascular diseases and cancer. It is suggested that the bioavailability and the metabolic pathways should be better understood before drawing any conclusions on the benefits of resveratrol to health. In CR15, Manna et al. reported that the anti-cancerogenic and anti-inflammatory effects of resveratrol are partially ascribed by the inhibition of the activation of a nuclear transcription factor (NP-kappa B and AP-1) and associated kinases. In CR16, Hung et al. reported that resveratrol was found to be a potent antiarrhythmic agent with cardioprotective properties in rats, possibly correlated with its antioxidant activity and upregulation of NO production. In CR17, Wallerath et al. reported about the effect of resveratrol on nitric oxide synthase expression (eNOS) in human endothelial cells. In conclusion, the stimulation of eNOS expression may contribute to the cardiovascular protective effects attributed to resveratrol. In CR18, Burns et al. examined the concentrations of resveratrol and its glycoside in grapes, peanuts, and Itadori tea, “which has long been used in Japan and China as a traditional herbal remedy for heart disease and stroke” [39]. The authors state that “for people who do not consume alcohol, Itadori tea may be a suitable substitute for red wine” [39]. In CR19, Wang et al. demonstrated for the first time, that resveratrol can cross the blood–brain barrier and exert protective effects against cerebral ischemic injury. In CR20, Baur and Sinclair comprehensively and critically reviewed in vivo data on resveratrol and considered its potential as a therapeutic for humans.

In CR21, Baur et al. discussed the improvement of health and survival of mice by resveratrol. The data “show that improving general health in mammals using small molecules is an attainable goal” [42]. In CR22, Lagouge et al. revealed that resveratrol improves mitochondrial function and protects against metabolic disease in mice. In CR23, Brown et al. examined the cancer chemoprotective activity in healthy volunteers and analyzed the pharmacokinetic effects after repeated doses of resveratrol. “The results suggest that repeated administration of high doses of resveratrol generates micromolar concentrations of parent and much higher levels of glucuronide and sulfate conjugates in the plasma. The observed decrease in circulating IGF-I and IGFBP-3 might contribute to cancer chemopreventive activity” [44]. In CR24, Cottart et al. reviewed the results of recent studies that investigated the pharmacokinetics, bioavailability, and toxicity of resveratrol in humans. As a result, resveratrol seems to be well tolerated with no marked toxicity, which provides further support for its use as a pharmacological drug in human medicine. In CR25, Pacholec et al. analyzed activation of SIRT1 by resveratrol and other structurally related compounds. The authors concluded that the tested compounds are not direct activators of SIRT1. In CR26, Patel et al. presented their study about the clinical pharmacology of resveratrol in colorectal cancer patients. The results suggest that daily p.o. doses of resveratrol at 0.5 or 1.0 g elicit anticancerogenic effects. The authors stated that “resveratrol merits further clinical evaluation as a potential colorectal cancer chemopreventive agent” [47]. In CR27, Chow et al. analyzed the effect of pharmacological doses of resveratrol on drug- and carcinogen-metabolizing enzymes of healthy volunteers. The authors concluded that resveratrol could modulate enzyme systems and thereby inhibit carcinogenesis. They suggest to consider lower doses of resveratrol for cancer prevention to minimize adverse metabolic drug interactions.

In CR28, Subbaramaiah et al. report on the inhibition of specific biochemical pathways in human mammary and oral epithelial cells. The authors state that their “data are likely to be important for understanding the anti-cancer and anti-inflammatory properties of resveratrol” [49]. In CR29, Howitz et al. found that resveratrol mimics caloric restriction of saccharomyces, resulting in increased DNA stability and extended lifespan of lower organisms. In CR30, Clement et al. reported on the anti-cancer activity of resveratrol by triggering apoptosis in human tumor cells. The authors state that their data highlight the chemotherapeutic potential of resveratrol. In CR31, Goldberg et al. reported on the absorption of three wine-related polyphenols (trans-resveratrol, catechin, and quercetin) in healthy males, measuring the polyphenols in blood serum and urine after oral consumption. The absorption of trans-resveratrol was discovered to be the most efficient. In CR32, Walle et al. found high absorption but very low bioavailability of oral resveratrol in humans. However, they state that accumulation of resveratrol and its metabolites in epithelial cells may still cause cancer-preventive effects. In CR33, Aggarwal et al. present an overview of preclinical and clinical studies investigating the role of resveratrol in the prevention and therapy of cancer. In CR34, Ray et al. studied the potential cardioprotective effects of resveratrol using isolated rat hearts. The results indicate “that resveratrol possesses cardioprotective effects which may be attributed to its peroxyl radical scavenging activity” [50]. In CR35, Bowers et al. reported on the biochemistry of resveratrol in connection with estrogen receptors, which seem to be important for the health effects of polyphenols. In CR36, Wu et al. reviewed the literature concerning the biosynthesis and presence of resveratrol in food groups and wines, the various reported health effects of resveratrol, and particularly the mechanism of its cardio-protective effect.

No additional cited references were identified within the 15 most referenced cited references (NCR ≥ 200). Therefore, we assume that we discovered the most impactful studies that are listed in Table 1 and Table 2.

## 4. Discussion

This bibliometric analysis deals with the health-related effects of resveratrol that occurs in wine and grapes. We employed the multidisciplinary database WoS due to the multidisciplinary nature of the topic. We analyzed the annual publication profile of the 3344 publications that were retrieved by our search query (see Appendix A). Publication activity increased until 2010 and decreased slightly afterwards. We analyzed frequently occurring author keywords and classified them into six groups: (1) beverage-related keywords, (2) compound-related keywords, (3) disease-related keywords, (4) effect-related keywords, (5) mechanism-related keywords, and (6) broader keywords. We analyzed and briefly discussed the most frequently cited references (i.e., key papers of the research topic analyzed) within the publication set.

Finally, we emphasize that we are not experts in the medical field and that more expert knowledge is needed for an in-depth interpretation of the results of our study. We presented a bibliometric method including the search query for retrieving the publication set and the appropriate analysis tools, i.e., VOSviewer and CRExplorer. Interested readers can study our RPYS results in greater detail via the interactive RPYS spectrogram. Alternatively, readers with access to WoS can retrieve the relevant publications or modify the search query (if desired) and download the search results for further analysis according to their own interests and needs.

The more recent publications had hardly any chance to accumulate higher citation counts until present. Therefore, the explanatory power of RPYS steadily decreases towards the present. The differentiation between higher and lower impact papers diminishes and key papers can hardly be identified based on their citation impact.

## 5. Conclusions

A comparatively large portion of the key papers revealed by the RPYS spectrogram and the N_TOP10 indicator takes a deliberative or positive attitude and reports about health effects of resveratrol. However, limited data in humans preclude drawing unambiguous conclusions on the health-related benefits of resveratrol. Conclusive evidence for resveratrol absorption by humans in biologically significant amounts and the mechanisms and metabolic pathways of the various potential health effects seem to be still lacking. In our analysis, we could not identify specific papers that imply a distinct change of direction of the ongoing scientific discourse. Moderate red wine consumption appears to bear the potential of benefitting health, whereas excessive alcohol consumption can induce liver cirrhosis and cancer. Evidently, further research and more in vivo studies are needed, before resveratrol (or red wine) sees clinical application as a pharmacological drug in human medicine.

## Figures and Tables

**Figure 1 ijerph-19-03110-f001:**
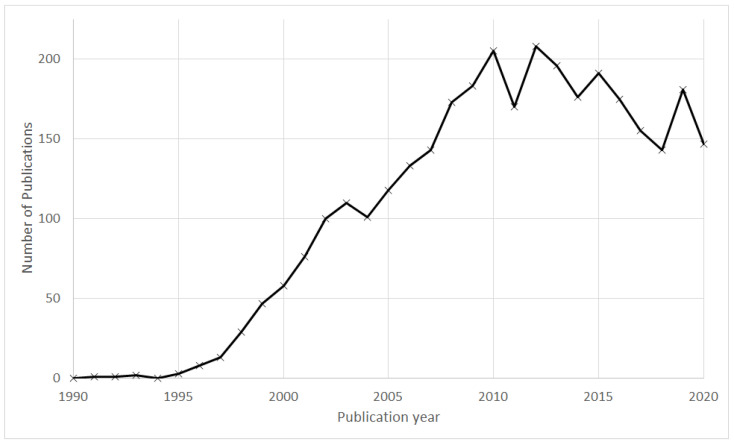
Annual publication profile of papers that deal with resveratrol in connection with wine and health.

**Figure 2 ijerph-19-03110-f002:**
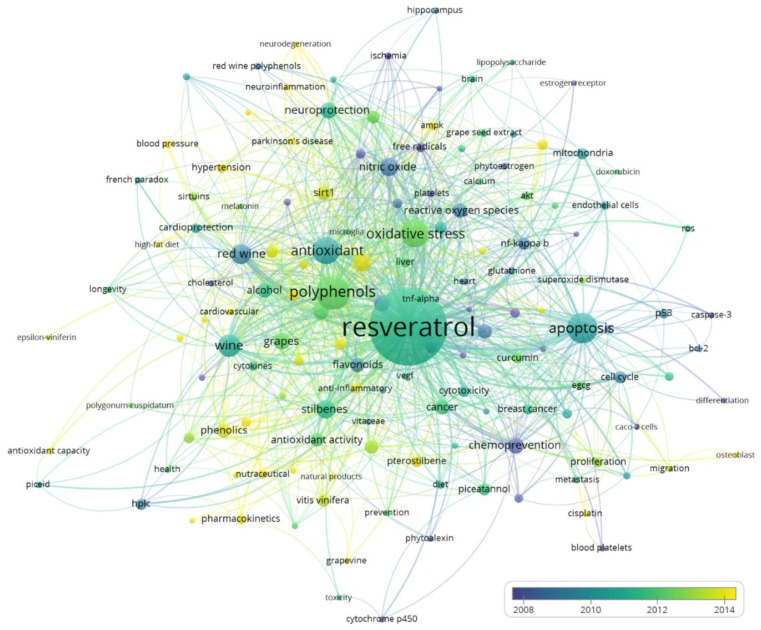
Semantic map of author keywords; clear synonyms (see Table A1) were grouped together; an interactive version is available at: https://s.gwdg.de/Q2BKwg, accessed on 8 February 2022.

**Figure 3 ijerph-19-03110-f003:**
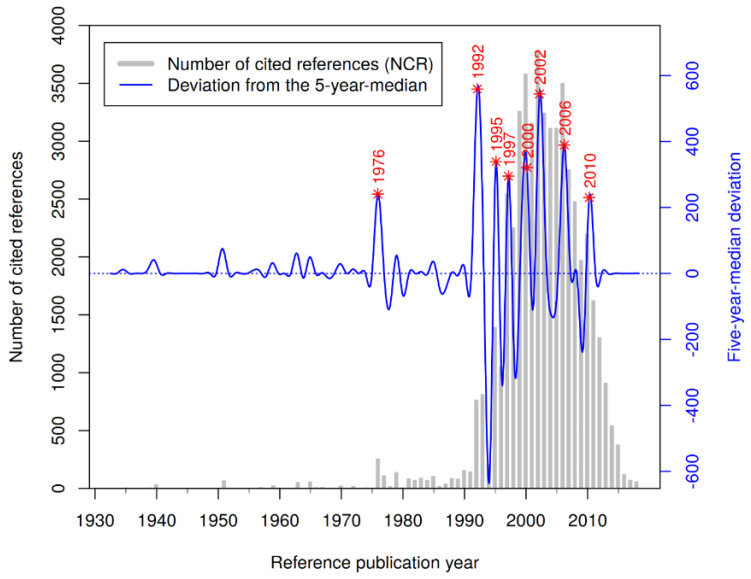
RPYS spectrogram of the wine- and health-related resveratrol research; outlier peaks are indicated by red stars (*) and marked with their RPY; an interactive version is available at: https://s.gwdg.de/i4qfNZ, accessed on 8 February 2022.

**Table 1 ijerph-19-03110-t001:** Most frequently cited references below the main peaks in Figure 3.

#CR	Cited Reference	RPY	NCR
1	Takaoka M., 1939, Nippon Kagaku Kaishi, V60, P1090, DOI 10.1246/nikkashi1921.60.1090 [4]	1939	12
2	Takaoka M.J., 1940, J. Fac. Sci. Hokkaido Imperi, V3, P1 [3]	1940	41
3	Langcake P., 1976, Physiol. Plant Pathol., V9, P77, DOI 10.1016/0048-4059(76)90077-1 [26]	1976	149
4	Renaud S., 1992, Lancet, V339, P1523, DOI 10.1016/0140-6736(92)91277-F [8]	1992	459
5	Siemann E.H., 1992, Am. J. Enol. Viticult., V43, P49 [1]	1992	200
6	Frankel E.N., 1993, Lancet, V341, P1103, DOI 10.1016/0140-6736(93)92472-6 [27]	1993	269
7	Frankel E.N., 1993, Lancet, V341, P454, DOI 10.1016/0140-6736(93)90206-V [28]	1993	120
8	Fitzpatrick D.F., 1993, Am. J. Physiol., V265, pH774 [29]	1993	87
9	Pace-Asciak C.R., 1995, Clin. Chim. Acta, V235, P207, DOI 10.1016/0009-8981(95)06045-1 [30]	1995	283
10	Bertelli A.A.E., 1995, Int. J. Tissue React., V17, P1 [31]	1995	166
11	Jang M.S., 1997, Science, V275, P218, DOI 10.1126/science.275.5297.218 [32]	1997	895
12	Gehm B.D., 1997, PNAS USA, V94, P14138, DOI 10.1073/pnas.94.25.14138 [33]	1997	289
13	Soleas G.J., 1997, Clin. Biochem., V30, P91, DOI 10.1016/S0009-9120(96)00155-5 [34]	1997	233
14	Fremont L., 2000, Life Sci., V66, P663, DOI 10.1016/S0024-3205(99)00410-5 [35]	2000	336
15	Manna S.K., 2000, J. Immunol., V164, P6509, DOI 10.4049/jimmunol.164.12.6509 [36]	2000	161
16	Hung L.M., 2000, Cardiovasc. Res., V47, P549, DOI 10.1016/S0008-6363(00)00102-4 [37]	2000	141
17	Wallerath T., 2002, Circulation, V106, P1652, DOI 10.1161/01.CIR.0000029925.18593.5C [38]	2002	189
18	Burns J., 2002, J. Agr. Food Chem., V50, P3337, DOI 10.1021/jf0112973 [39]	2002	111
19	Wang Q., 2002, Brain Res., V958, P439, DOI 10.1016/S0006-8993(02)03543-6 [40]	2002	108
20	Baur J.A., 2006, Nat. Rev. Drug. Discov., V5, P493, DOI 10.1038/nrd2060 [41]	2006	427
21	Baur J.A., 2006, Nature, V444, P337, DOI 10.1038/nature05354 [42]	2006	302
22	Lagouge M., 2006, Cell, V127, P1109, DOI 10.1016/j.cell.2006.11.013 [43]	2006	197
23	Brown V.A., 2010, Cancer Res., V70, P9003, DOI 10.1158/0008-5472.CAN-10-2364 [44]	2010	75
24	Cottart C.H., 2010, Mol. Nutr. Food Res., V54, P7, DOI 10.1002/mnfr.200900437 [45]	2010	66
25	Pacholec M., 2010, J. Biol. Chem., V285, P8340, DOI 10.1074/jbc.M109.088682 [46]	2010	56
26	Patel K.R., 2010, Cancer. Res., V70, P7392, DOI 10.1158/0008-5472.CAN-10-2027 [47]	2010	55
27	Chow H.H.S., 2010, Cancer Prev. Res., V3, P1168, DOI 10.1158/1940-6207.CAPR-09-0155 [48]	2010	48

**Table 2 ijerph-19-03110-t002:** Cited references that were cited very frequently in at least 15 different citing years (i.e., N_TOP10 ≥ 15) that are not included in Table 1.

#CR	Cited Reference	RPY	NCR	N_TOP10
28	Subbaramaiah K., 1998, J. Biol. Chem., V273, P21875, DOI 10.1074/jbc.273.34.21875 [49]	1998	216	17
29	Howitz K.T., 2003, Nature, V425, P191, DOI 10.1038/nature01960 [51]	2003	275	17
30	Clement M. V., 1998, Blood, V92, P996 [52]	1998	223	16
31	Goldberg D. A., 2003, Clin. Biochem., V36, P79, DOI 10.1016/S0009-9120(02)00397-1 [53]	2003	156	16
32	Walle T., 2004, Drug Metab. Dispos., V32, P1377, DOI 10.1124/dmd.104.000885 [54]	2004	259	16
33	Aggarwal B.B., 2004, Anticancer Res., V24, P2783 [55]	2004	202	16
34	Ray P.S., 1999, Free Radical Bio. Med., V27, P160, DOI 10.1016/S0891-5849(99)00063-5 [50]	1999	158	15
35	Bowers J.L., 2000, Endocrinology, V141, P3657, DOI 10.1210/en.141.10.3657 [56]	2000	114	15
36	Wu J.M., 2001, Int. J. Mol. Med., V8, P3 [6]	2001	105	15

## Data Availability

The data are property of Clarivate Analytics and are available from them using our search query.

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
