# Peer review of "On Health Effects of Resveratrol in Wine"

_ijerph, 2022, doi:10.3390/ijerph19053110_

Round 1

Reviewer 1 Report

Suggestions

The manuscript is very good proposal and have been focused on a review paper to verify, in literature, the real health effects of wine and grapes resveratrol molecule.

Line 25-26: the authors must specify what the word “variety” refers to in the sentence; also, the statement must be referenced

Please, review the sentence “Error! Reference source not found”, founded in different parts of manuscript

The Table 1 should not be separated into two pages on the manuscript, please check the table

Line 225-226: in the sentence “vitis vinifera and other vitaceaes”, please write in italics and capital letters

Line 231: the authors must specify what the word “variety” refers to in the sentence

Line 236: please standardize the way of writing in vitro and in vivo, and preferably in italics and without hyphen

The manuscript showed a very good search strategy in database and the conclusion are supported by different references used in this study.

Author Response

R1-P1) The manuscript is very good proposal and have been focused on a review paper to verify, in literature, the real health effects of wine and grapes resveratrol molecule.
Ad R1-P1) Thank you!
R1-P2) Line 25-26: the authors must specify what the word “variety” refers to in the sentence; also, the statement must be referenced
Ad R1-P2) We have replaced the term “variety” with “wine grape variety“ to make the reference clear. Furthermore, we have referenced CR5 from Table 1 in lines 25-26 because it relates the wine grape variety with resveratrol concentration.
R1-P3) Please, review the sentence “Error! Reference source not found”, founded in different parts of manuscript
Ad R1-P3) We have replaced the broken references with the proper ones.
R1-P4) The Table 1 should not be separated into two pages on the manuscript, please check the table
Ad R1-P4) We have moved parts of the text to make the figures and tables fit better on the pages.
R1-P5) Line 225-226: in the sentence “vitis vinifera and other vitaceaes”, please write in italics and capital letters
Ad R1-P5) Done.
R1-P6) Line 231: the authors must specify what the word “variety” refers to in the sentence
Ad R1-P6) We have consistently replaced the term “variety” with “wine grape variety“.
R1-P7) Line 236: please standardize the way of writing in vitro and in vivo, and preferably in italics and without hyphen
Ad R1-P7) Thank you for catching this inconsistency. We standardized both terms to the form without hyphen in italics.
R1-P8) The manuscript showed a very good search strategy in database and the conclusion are supported by different references used in this study.
Ad R1-P7) Thank you!

Reviewer 2 Report

The article showed the analysis of 3344 publications related to the health effects of  resveratrol in wine and grapes. They suggested that they could not identify specific publications that provide a distinct change of direction however moderate red wine consumption may have the potential of being health promoting while excessive alcohol consumption can induce liver cirrhosis and cancer.

The article has valuable information and I suggest to fix following minor revisions before publish.

 Introduction has some places that needs to cite references. For example, Lines 25, 29, 35, 37 : References were not cited

Line 144, what do you mean by all document types? Explain.

Add the most recent references on resveratrol  as well.

I would encourage authors to discuss more about the results to add contents to discussion and conclusions section.

Author Response

R2-P1) The article showed the analysis of 3344 publications related to the health effects of resveratrol in wine and grapes. They suggested that they could not identify specific publications that provide a distinct change of direction however moderate red wine consumption may have the potential of being health promoting while excessive alcohol consumption can induce liver cirrhosis and cancer.
The article has valuable information and I suggest to fix following minor revisions before publish.
Ad R2-P1) Thank you!
R2-P2) Introduction has some places that needs to cite references. For example, Lines 25, 29, 35, 37 : References were not cited
Ad R2-P2) We have cited additional references as proposed by the reviewer.
R2-P3) Line 144, what do you mean by all document types? Explain.
Ad R2-P3) We provided examples of the other document types that are contained in our data set.
R2-P4) Add the most recent references on resveratrol  as well.
Ad R2-P4) We cited a recent review regarding the pharmacological literature connected to resveratrol in the introduction (lines 43-44).
R2-P5) I would encourage authors to discuss more about the results to add contents to discussion and conclusions section.
Ad R2-P5) We prefer to keep the discussion and conclusions short and to the point. We lack medical knowledge for a deeper discussion.